# Correlations Between Mammographic Breast Density and Outcomes After Neoadjuvant Chemotherapy in Patients with Locally Advanced Breast Cancer

**DOI:** 10.3390/cancers17132214

**Published:** 2025-07-01

**Authors:** Veenoo Agarwal, Lisa Spalding, Hilary Martin, Ellie Darcey, Jennifer Stone, Andrew Redfern

**Affiliations:** 1Fiona Stanley Hospital, Murdoch, WA 6150, Australia; hilary.martin@health.wa.gov.au; 2School of Medicine, The University of Western Australia, Crawley, WA 6009, Australia; ellie.darcey@uwa.edu.au (E.D.); jennifer.stone@uwa.edu.au (J.S.); 3Harry Perkins Institute of Medical Research, Robin Warren Drive, Murdoch, WA 6150, Australia; lisa.spalding@perkins.org.au

**Keywords:** mammographic density, pathological response, breast cancer

## Abstract

This study investigates whether the amount of dense breast tissue on the mammograms of patients with early breast cancer affects how well chemotherapy given before surgery shrinks their cancer and whether they survive their cancer. We measured mammographic breast density along with the rate of disappearance of cancer for the time of surgery, cancer return rates and survival in 127 women undergoing chemotherapy before surgery for early breast cancer. In the whole patient group, no significant differences were seen. However, when we looked at the effects of weight, obese patients with dense breasts had worse outcomes than obese patients with non-dense breasts; no tumours were microscopically eliminated versus 28%, and rates of cancer return and deaths doubled. Non-obese patients showed no such pattern. These findings suggest that breast density and high body weight together may hinder chemotherapy success, and that research on better treatments for these patients is needed.

## 1. Introduction

Mammographic breast density (MBD) represents a white radiographic appearance of dense epithelial and stromal breast tissue as opposed to a dark appearance of non-dense (fatty) breast tissue on a mammogram. At a population level, high MBD is the most substantial risk factor for breast cancer (BC) after age and BRCA1/2 genetic mutation carrier status [1,2]. High MBD is common, with 43% of women aged 40–74 estimated to have heterogeneously dense or extremely dense breasts [3]. As well as being linked with increased BC risk, high MBD has been associated with increased local BC recurrence after breast-conserving surgery for early BC (tumour confined to the breast and axillary lymph nodes), with five out of six studies showing a significant increase in local relapse with high MBD relative to low MBD [4]. However, no increase in distant relapse was seen in these cohorts. In contrast, a reduction in MBD during endocrine therapy has been shown to strongly correlate with a reduced risk of relapse across multiple studies [5].

Chemotherapy is commonly deployed alongside surgery in early BC for well-established improvements in both relapse-free survival (RFS) and breast cancer-specific survival (BCSS) [6]. The backbone chemotherapy agents of most regimens include anthracyclines which inhibit topoisomerase II, leading to DNA strand breaks and taxanes which interfere with cellular microtubules, both during division. Increasingly, these are given neoadjuvantly (prior to surgery), which can both facilitate a more conservative operation by reducing tumour size, as well as guide further post-surgical treatment in select patient groups [7,8]. The achievement of a pathological complete response (pCR), the complete microscopic disappearance of all invasive tumour in breast and lymph node surgical specimens after neoadjuvant treatment, is correlated with improved long-term outcomes, particularly in triple-negative, HER2-positive and some luminal B cancers [9].

Although pCR for specific BC subtypes correlates with improved RFS and BCSS in large general cohorts, this does not necessarily hold for all individual-specific groups, or specific treatments. For example, the anti-angiogenesis agent bevacizumab, when added to NAC, significantly contributes to higher pCR rates, but not to improved survival [10]. More broadly, a recent meta-analysis found pCR to be a poor surrogate for longer-term outcomes across general cohorts in neoadjuvant BC trials [11]. In particular, MBD, as it describes a local biological breast tissue phenomenon, could impact the chemosensitivity of the primary breast tumour without impacting the response of micro-metastases in distant sites, the presumptive driver of altered survival.

In the neoadjuvant setting, a number of studies have explored potential links between high MBD and chemotherapy resistance, with somewhat mixed results [12,13,14,15,16,17]. Both the earliest [12] and the most recent studies [16] found no link between MBD and chemotherapy response in univariate analysis. Two further studies have linked high MBD with significantly lower pCR at surgery after neoadjuvant chemotherapy (NAC) relative to low MBD in multivariable analysis [13,14,15]. In contrast, a fifth study found a different pattern with the highest pCR rates in cases of medium MBD, and lower pCR rates in the quartiles with the lowest and highest MBD [17]. To date, only the study of Castaneda et al. [12] reported any survival data. All other studies confined reporting to correlations between MBD and pCR alone. As pCR is only useful insofar as it can be a surrogate for survival, and as variations in MBD could affect this association, we set out to further explore the relationship between high MBD and chemotherapy response, as well as to assess whether any identified relationship extends to differing relapse rates and survival.

This study aimed to validate the relation between MBD and pCR in patients undergoing neoadjuvant chemotherapy (NAC) for early BC at a single tertiary centre, Royal Perth Hospital (RPH), Western Australia, between 2003 and 2014. For a secondary measure of locoregional response we measured primary tumour shrinkage. As assessing response in the breast by imaging is not routine in clinical practice but the majority of patients undergo regular clinical examination, we also set out to assess the relation between MBD and tumour reduction by clinical complete response, as well as relapse-free (RFS) and BC-specific survival (BCSS). Given the apparent influence of body mass index (BMI) on the relationship between MBD and pCR in previous studies [14,15], we also set out to explore any interaction between BMI, MBD and chemoresponse.

## 2. Materials and Methods

### 2.1. Patient Population

Patients undergoing NAC for non-metastatic BC at Royal Perth Hospital (RPH), a tertiary hospital in Western Australia, between January 2003 and December 2014 were identified from the institutional Medical Oncology Department database. All patients had no evidence of metastatic disease on initial staging, had at least four cycles of neoadjuvant anthracycline and/or taxane-based chemotherapy and underwent curative-intent tumour resection following NAC completion. All BC sub-types were included. Patients with human epidermal growth factor receptor 2 (HER2)-amplified tumours receiving neoadjuvant trastuzumab with chemotherapy were also included, but patients who received neoadjuvant endocrine therapy were not. The exclusion criteria were previous invasive malignancy in the past five years, previous invasive BC at any previous time, inflammatory BC, bilateral BC and patients without eligible contralateral mammograms (see definition below).

Patient demographics, including age and BMI, were extracted from the hospital database and individual medical records. For the analysis comparing the impact of MBD on chemoresponse between obese and non-obese individuals, obesity was defined as BMI > 30 kg/m^2^. Tumour pathology data were obtained from pre-NAC biopsy and post-operative surgical excision anatomical pathology reports. The BC sub-type was established from the pre-NAC tumour grade, estrogen receptor (ER), progesterone receptor (PR) and HER2 immunohistochemistry. Receptor statuses were considered positive as per the latest ASCO/CAP guidelines [10,17]. Where receptor status was not available for pre-operative tissue, post-operative test results were used.

Treatment delivered was established from medical records with pharmacy data accessed additionally for incomplete data. Pre- and post-NAC primary tumour sizes and lymph nodes, where relevant, were based on clinical examination data. Clinical tumour size was calculated by the multiplication of the longest tumour diameter by the perpendicular diameter, usually measured by callipers. Degree of clinical response was calculated for a patient by expressing the post-chemotherapy size as a percentage of the pre-chemotherapy size. Clinical complete response (cCR) required the complete resolution of the primary breast mass and any enlarged lymph nodes on examination. Clinical partial response (cPR) was calculated in line with the WHO response criteria [18] (Miller 1981), which require a 50% reduction in tumour size in two perpendicular dimensions. For the purposes of our study, cPR was defined as a 75% reduction in the primary tumour area, representing the net product of 50% reductions in two perpendicular dimensions. The term clinical stable disease (SD) was assigned to tumours with less than 75% shrinkage but less than 25% growth. No patient in this study cohort experienced tumour growth on treatment.

### 2.2. Mammographic Breast Density Assessment

Mammography images taken at or up to 12 months prior to diagnosis were collated and used for MBD assessment. Film mammograms (pre-2008) were retrieved and digitized using a high-powered scanner; copies of full-field digital mammograms (2008 onwards) were retrieved from the Western Australian Picture archiving and communication system (WA PACS). Only craniocaudal (CC) views of the contralateral breast were assessed to avoid confounding density appraisal by the presence of either tumour or post-operative scarring on the affected side. Patients were considered ineligible if CC view mammograms were missing or were post-surgical, of the ipsilateral breast, or of insufficient image quality for reliable density assessment.

MBD assessment, including total dense area (DA), percentage dense area (PDA) and non-dense area (NDA), was performed using Cumulus software Version 1.0 (Sunnybrook Health Sciences Centre, Toronto, ON, Canada).

We found that the range of PDAs for film mammograms was higher than for digital images, with median PDAs of 20% and 11%, respectively. When we came to assess MBD as a dichotomous variable, film and digital mammography cohorts were separated and patients assigned to high- or low-MBD patients based on their PDA relative to the median PDA within their cohort. Low-MBD patients and high-MBD patients from each mammogram type cohort were then combined to give two approximately equal groups of high- and low-MBD individuals for assessment. The median was chosen as a cut-point, as this reflects current clinical assessments whereby the BIRADs MBD scoring system [19] (D-orsi et al., 2003) designates the upper two categories, heterogeneously dense and extremely dense breasts, as high breast density, which corresponds to approximately 50% of women in our region.

## 3. Results

### 3.1. Included Population

A total of 238 patients were identified as receiving appropriate NAC followed by curative intent surgery in the given period. For 71 patients, no suitable contralateral mammogram was available, and for 40, the derived digital film was not assessable by Cumulus. This left 127 patients available for analysis, comprising 68 with film mammography and 59 with digital images (Figure 1). For the 111 ineligible patients, more patients were from the pre-2008 film mammography group (*n* = 63) than the 2008-onward digital mammogram group (*n* = 48) due to older and physically archived images being more likely to be untraceable than digital images. However, no clinically or statistically significant differences in personal and tumour demographics or treatment were observed for these excluded patients.

Patient demographic and tumour features were similar between included film and digital mammogram cohorts, with an increase in HER2-positive tumours in the digital cohort being the only significant difference (*p* = 0.003, Table 1). This likely reflects the shift in protocols and practice from selecting patients for NAC based on primary tumour size in the earlier cohort to selection based on tumour biology in the later cohort. In more recent years, NAC has been routinely incorporated into HER2-positive early breast cancer for all but small tumours. The treatment delivered to the two cohorts was comparable (Table 2), with 77.6 and 75.9% of film and digital mammography patients receiving at least six cycles of neoadjuvant therapy, respectively (*p* = 0.81). Notably, few HER2-positive patients in either film or digital groups received neoadjuvant trastuzumab (9.1 vs. 13.6%, *p* = 0.798).

### 3.2. Response Rates and Survival

#### 3.2.1. Overall Population Outcomes

For the whole cohort, clinical complete response (cCR) was 49.1% and pathological complete response (pCR) was 21.3%. The median follow-up was 9.3 years; 13.5 years for film mammography and 5.6 years for digital mammography. A total of 33.9% of patients had experienced disease relapse, and 28.3% had died of BC by the study censor date. pCR in the whole population correlated with both improved RFS (*p* = 0.002) and BCSS (*p* = 0.0035). Patients with cCR also had improved RFS compared to those with cPR (*p* = 0.0134) or those with stable disease (SD) (*p* = 0.0037). cCR also correlated with improved BCSS relative to those with SD (*p* = 0.0057).

#### 3.2.2. Interaction of Mammographic Breast Density with Response and Survival

Considering the associations of MBD as a continuous variable and BC outcomes (Table 3, Figure 2), the mean PDA was significantly lower in those experiencing a cCR than those not (16.8 vs. 22.2%, *p* = 0.048), but did not significantly vary between those experiencing a pCR or not (17.9 vs. 19.2%, *p* = 0.375). In contrast, the mean PDA was significantly higher in relapsing compared to non-relapsing patients (22.4 vs. 17.1%, *p* = 0.041), with a similar but non-significant trend for distant relapse (22.1 vs. 17.6%, *p* = 0.071), although mean PDA did not vary by BC deaths (21.2 vs. 18.0, *p* = 0.144).

Considering MBD as a dichotomous variable, as seen for mean density, patients with low MBD had a higher rate of cCR than those with high MBD (58.5 vs. 40.0%, *p* = 0.027) (Table 4). No significant difference was seen for pCR between low- and high-MBD groups (25.0 vs. 17.5%, *p* = 0.150). Although relapse, distant relapse and BC-specific mortality rates were all numerically higher in high- relative to low-MBD patients, no statistically significant differences were seen for any outcome event (Table 4). This was supported by the Kaplan–Meier plots showing risk over time (Figure 3). Although a moderate separation was seen for RFS (Figure 3a), this did not reach significance (*p* = 0.098) Similarly, BCSS did not vary by MBD category (Figure 3b).

Considering potential confounders, there was no difference between tumour demographics or biology between the high and low mammographic density cohorts. In particular, HER2 status, the biggest differential between film and digital mammography groups, did not vary, with 30.6 and 29.2% of patients in low and high breast density breast groups being HER2-positive.

#### 3.2.3. The Impact of Body Mass Index

Specific attention was paid to the impact of BMI given that this factor exerted a significant effect in the two studies showing an association between MBD and chemotherapy response published to date [10,11], whereas negative studies generally made no adjustment for BMI. A moderate relationship was seen between BMI and MBD when these were considered as continuous variables in the whole population (Figure 4).

An exploratory analysis looked at the effect of MBD on pCR, RFS and BCSS in different BMI categories. Obese patients had significantly lower response rates if they had high MBD relative to low MBD, with pCR rates being 0% vs. 28.1% (*p* = 0.036), relapse rates 56% vs. 28% (*p* = 0.063) and breast cancer deaths 56 vs. 28%, (*p* = 0.071) in high- and low-MBD cohorts, respectively (Figure 5). In contrast, no differences in non-obese patients were seen for either pCR (20.0 vs. 22.2%, *p* = 0.409), relapse rates (36 vs. 33%, *p* = 0.408) or breast cancer deaths (26 vs. 22%, *p* = 0.357) between high- and low-MBD cohorts.

To exclude confounding factors in the relationship between high BMI and high BMD, we compared features of tumours by intersecting MBD and BMI categories (Appendix A). Comparing the patients with high MBD and high BMI to all others, notably, the former have significantly more T3-4 tumours. Obese patients regardless of MBD category had numerically more involved nodes, but numerically less high-grade cancers. Patients with high MBD regardless of BMI are also younger, in keeping with the well-established correlation between higher MBD and younger age seen across multiple populations. Low numbers in individual cohorts precluded a formal meta-analysis.

## 4. Discussion

In this study, we explore the correlation between MBD and both clinical and pathological complete response as well as longer-term outcomes after NAC. Considering MBD specifically as a continuous variable, a higher mean PDA was significantly associated with inferior cCR and RFS, but not with pCR or BCSS, although the numerical directions of these relationships were the same. The lack of statistical significance in some measures could be attributed to insufficient events (cCR events and relapse events are higher than pCR and breast cancer death events). Alternatively, it could be that high MBD, and the underlying stroma this represents, reduce the ability of chemotherapy to kill bulk cell populations relative to low MBD lowering cCR, but not to eradicate all cancer clones in responders, leading to a similar pCR. Additionally, as cCR is a clinical measure, high MBD may reduce the clinician’s ability to confirm the disappearance of a tumour leading to a spuriously low cCR.

Turning to MBD as a dichotomous variable, when dividing the population at the median PDA into high- and low-MBD groups, cCR was significantly lower and pCR, RFS and BCSS were all numerically but non-significantly worse in high MBD. Again, although this may be a product of lower events for the latter outcome measures, it may be that high MBD reduces general bulk cytotoxicity but has less of an impact on the eradication of all cancer cell clones, including the micro-metastatic disease that is usually responsible for relapse.

Our patient cohort was divided into two distinct groups: an earlier population diagnosed using film mammography, and a later population diagnosed using digital mammography. The earlier film mammography population had significantly higher mean and median total DA and PDA. Although this appears most likely to relate to differences generated by the film technique, the higher level of obesity in the later cohort, which has an identified correlation with lower MBD [20], may have contributed.

There are now five studies correlating neoadjuvant chemo-response and MBD, in a total of 2091 patients [12,13,14,15,16,17]. All studies had relatively high-risk populations, with lymph node involvement ranging from 51 to 81%, and comparable rates of triple-negative and HER2-positive tumours. Where recorded, 70 to 90% received comprehensive chemotherapy including both anthracycline and taxane components. The earliest study, from Peru, showed no correlation between MBD, as measured by the BI-RADS method and pCR. This is the only other study to report basic survival data with no difference in median PFS or overall survival by MBD category [12]. The most recent study, from Ireland, also found no significant link between MBD and pCR, with no longer-term outcomes reported [16]. Another recent study from Italy found intermediate MBD categories by the BI-RADs method to have the highest pCR, significantly so in multivariable analysis, which included BMI among other factors, with lower pCRs at the extremes of MBD. Again, no survival outcomes were reported [17]. The Swedish and Saudi Arabian studies showed the most comparable results with our own, finding significant or borderline-significant links between higher BMD and lower pCR, although in both cases only after multivariable analysis accounting for BMI. Again, neither reported longer-term outcomes [14,15].

Considering underlying reasons for this heterogeny of outcomes, the first difference of note is the method of density measurement employed. Our own study and that of Elsamany and colleagues utilized methods that produce a continuous percent score of density, allowing us to assess the influence of density on outcome as both a continuous and categorical variable. We used only the contra-lateral breast film to avoid confounding the assessment due to either the presence of tumour or scar tissue. All other studies employed the BIRADS measurement system that produces a categorical MBD score [21] and uses either data from both breasts or the denser breast, such that the increased density of the tumour itself could confound the results.

A further possible explanation for some of the variation between studies is provided by our hypothesis-generating finding that outcomes only differed by MBD in obese patients. Notably, no obese patient with high MBD achieved a pCR. Our cohort and that of the Saudi Arabian study had higher proportions of obese patients (35 and 47%, respectively) than the Peruvian study (31%). Swedish and Italian studies did not report rates of obesity. A large meta-analysis of 82 trials found high BMI to correlate with poor breast cancer outcomes [22].

A number of biological mechanisms could underpin chemoresistance produced by the interaction between BMI and MBD. In obesity, fat becomes hypoxic, resulting in the dysregulation of cytokine and growth factor output [23]. The factors released, including from hypoxic tissue, have the capacity to drive tumour progression and treatment resistance [24]. Consequently, a relationship between obesity, MBD and chemoresistance could arise from the tumour-promoting systemic endocrine and growth factor milieu that arises in obesity, combined with the promotion of chemoresistance driven by features of high-density local stroma.

Turning to such potential mechanisms local to the breast tissue itself whereby dense stroma could contribute to chemoresistance, the first hypothesis to explore is that lower MBD simply associates with tumour types that are intrinsically more responsive to chemotherapy. However, a comprehensive review has demonstrated no correlation between MBD and basic tumour biology [4] making this an unlikely explanation. Alternatively, differing immune cell infiltrates between high- and low-MBD areas could play a role, with a significant increase in macrophages, in particular, being demonstrated in high- compared with low-MBD epithelium [25]. In other studies, overall macrophage numbers have been correlated with poorer chemo-response in breast cancer patients [26]. Finally, epithelial–mesenchymal transition (EMT) has been correlated with chemoresistance [27] and subsequent accelerated tumour progression post-therapy in breast cancer [28]. Mesenchymally transitioned cells have been identified in breast cancer cell lines cultured in artificial high-density collagen and in breast tumours in women with high MBD [29].

Taken together, we hypothesize that in high MBD, dense collagen combined with macrophage infiltrates may trigger EMT and other mechanisms to drive chemoresistance, promoted by the systemic hypoxic pro-proliferative environment particularly found in obese patients.

A strength of our study includes conduct of the first assessment of the interaction of MBD with clinical response and long-term treatment outcomes. The inclusion of analysis by BMI is also novel, although this requires validation. Weaknesses include the relatively low case number, which precluded analysis of the predictive import of MBD in breast cancers of differing biological sub-type. Furthermore, for the patients with high BMI and high MBD, the higher rate of larger tumours with more frequent lymph node involvement would tend to adversely affect outcomes, although the lower number of high-grade tumours would be expected to compensate to a degree. Again, validation in a larger cohort is required. Additionally, the low rate of use of HER2-targeted agents in the HER20positive cohort makes these results relevant for contemporary HER20-positive patients, where such targeting is routinely uncertain. The heterogeneity of measured density ranges between film and digital mammography groups also makes the validity of the combined analysis uncertain.

Looking forward, we are expanding our digital mammography cohort to confirm or refute the relationship between MBD and the outcomes observed, including the modulating impact of obesity. We also plan to assess the correlation between EMT and EMT transcription factors and both MBD and chemoresponse. Considering the field more generally, further analysis of existing published cohorts that have explored MBD and chemoresponse correlations to compare quantitative methods such as Cumulus and qualitative methods such as BIRADS, as well as exploring the modifying influence of obesity on relationships, may contribute.

## 5. Conclusions

This study set out to examine the relationship between MBD, chemotherapy response and long-term outcomes in women with early breast cancer treated with neoadjuvant chemotherapy. While higher MBD was linked to lower rates of clinical complete response, no significant differences were noted in survival outcomes across the complete cohort. However, among obese patients with high MBD, poorer responses to chemotherapy and worse relapse and survival rates were observed, a pattern not seen in non-obese patients. These findings suggest that the combination of high breast density and obesity may negatively impact treatment success. Further research is needed to better understand the underlying biology of this interaction and to guide more personalized treatment strategies for these higher-risk chemo-resistant patients.

## Figures and Tables

**Figure 1 cancers-17-02214-f001:**
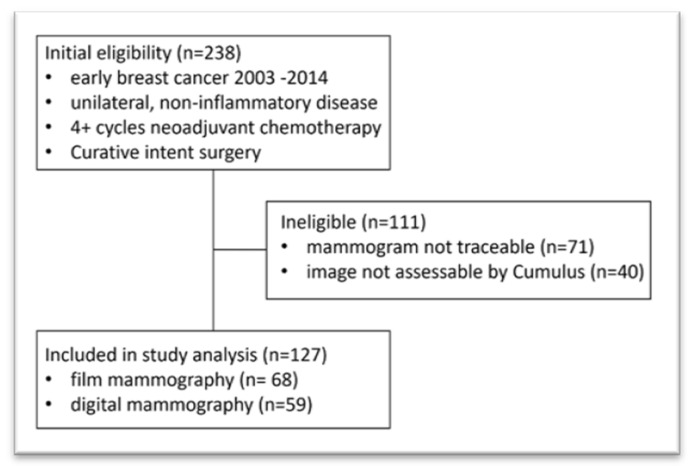
Patient CONSORT diagram.

**Figure 2 cancers-17-02214-f002:**
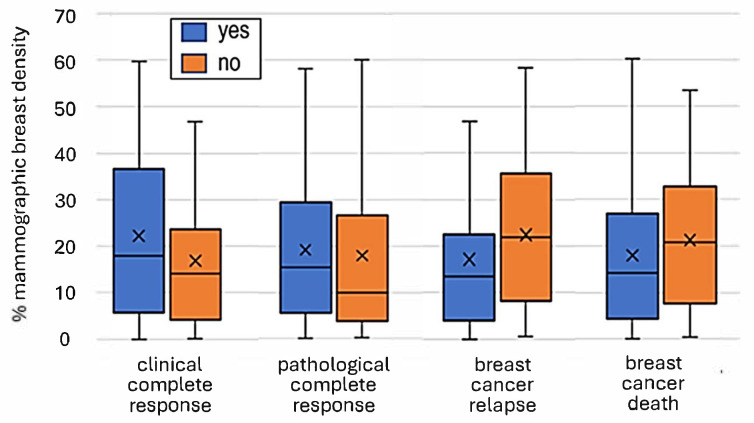
Relationship between percentage mammographic breast density and clinical complete response, pathological complete response, breast cancer relapse and breast cancer death. X denotes mean density for a cohort.

**Figure 3 cancers-17-02214-f003:**
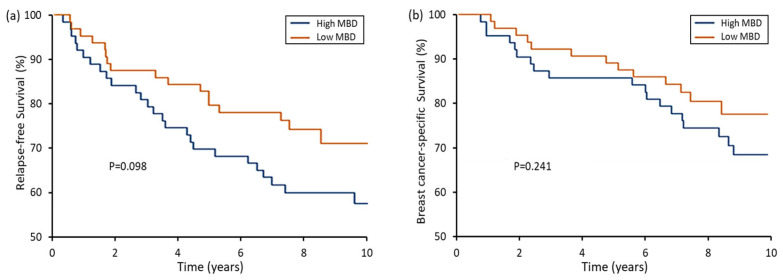
Kaplan–Meier curves for relationship between MBD and (**a**) relapse-free survival or (**b**) breast cancer-specific survival.

**Figure 4 cancers-17-02214-f004:**
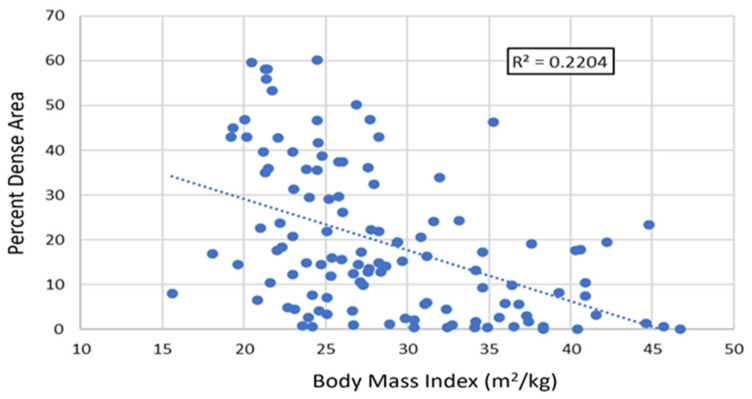
Interaction of body mass index with mammographic percent dense area as continuous variables.

**Figure 5 cancers-17-02214-f005:**
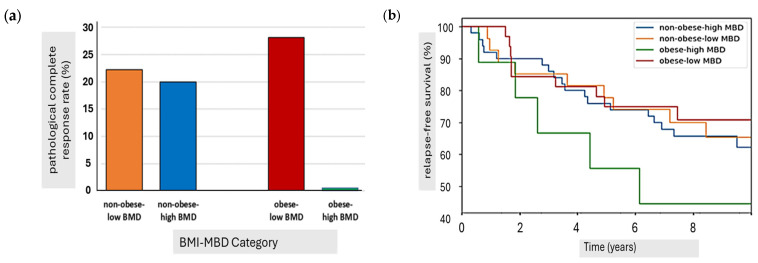
Interaction of mammographic breast density and BMI, demonstrating (**a**) pCR rates and (**b**) RFS for cohorts by BMI and breast density category.

**Table 1 cancers-17-02214-t001:** Individual and tumour demographics of patients, including mammogram type.

		All Patients		Patients Divided by Mammogram Type				
				Film		Digital		
		*n*	%	*n*	%	*n*	%	*p* diff
**Total**		127	(100.0)	68	(53.4)	59	(46.6)	
**Age**	**Range (yrs)**	28–97		28–97		29–92		0.865 **
	**Median (yrs)**	48		47		51		
**BMI**	**Range (kg/m^2^)**	15.6–46.7		19.2–44.8		15.6–46.7		0.281 **
	**Median (kg/m^2^)**	27.1		26.6		28.7		
	**<18.5**	2	(1.7)	0	(0)	2	(3.6)	0.390 **
	**18.5–24.9**	40	(33.9)	26	(41.3)	14	(25.5)	
	**25.0–29.9**	35	(29.7)	17	(27.0)	18	(32.7)	
	**30.0 +**	41	(34.7)	20	(31.7)	21	(38.2)	
**Size**	**Range (mm)**	3–140		3–140		10–135		0.072 **
	**Median (mm)**	60		63		52		
**T stage**	**T1**	3	(2.5)	2	(3.0)	1	(1.9)	0.44 *
	**T2**	48	(40.7)	23	(34.8)	25	(48.1)	
	**T3**	48	(40.7)	28	(42.4)	20	(38.4)	
	**T4**	19	(16.1)	13	(19.7)	6	(11.5)	
**LN positive**	**No**	57	(48.7)	31	(47.7)	26	(50.0)	0.950 ***
	**Yes**	60	(51.3)	34	(52.3)	26	(50.0)	
**Grade**	**1**	10	(8.0)	7	(10.4)	3	(5.2)	
	**2**	57	(45.0)	32	(47.8)	25	(43.1)	0.390 ***
	**3**	58	(46.4)	28	(41.8)	30	(51.7)	
**Receptors**	**ER/PR positive**	84	(66.1)	45	(66.2)	39	(66.1)	1.000 ***
	**HER2 positive**	36	(28.8)	11	(16.7)	25	(42.4)	0.003 ***
**Sub-type**	**Lum A**	47	(37.3)	29	(43.3)	18	(30.5)	0.110 ***
	**Lum B**	36	(28.6)	15	-22.4	21	(35.6)	
	**TNBC**	28	(22.2)	17	(25.4)	11	(18.6)	
	**HER2 enriched**	15	(11.9)	6	(9.0)	9	(15.2)	

* statistical comparison T3 + T4 tumours combined to T1 + T2 tumours combined. ** statistical comparison for age, BMI and size by paired *t*-test. *** other statistical comparisons by Chi square test.

**Table 2 cancers-17-02214-t002:** Treatment delivered to patients including by mammogram type cohort.

			All Patients		Patients Divided by Mammogram Type				
					Film		Digital		*p* diff
			*n*	%	*n*	%	*n*	%	Chi sq
**Total patients**			127	(100.0)	68	(53.4)	59	(46.6)	
**Neoadjuvant Treatment**									
**Chemotherapy**	**Median cycles**		6		6		6		
	**Mean cycles**		5.7		5.6		5.9		
	**6+ cycles**		97	(77.6)	53	(79.1)	44	(75.9)	0.814
	**Anthracycline**		13	(10.4)	5	(7.5)	8	(13.8)	-
	**Taxane**		5	(4.0)	1	(1.5)	4	(6.9)	
	**Anthracycline + Taxane**		107	(85.6)	61	(91.0)	46	(79.3)	0.089
**HER2 targeted**	**Trastuzumab**		4	(11.4)	1	(9.1)	3	(13.6)	0.798
**Combined (Neo)adjuvant Treatment**									
**Chemotherapy**	**Median cycles**		6		6		6		
	**Mean cycles**		6.4		6.2		6.6		
	**6+ cycles**		110	(88.0)	60	(89.6)	50	(86.2)	0.877
	**Anthracycline**		4	(3.2)	2	(3.0)	2	(3.4)	-
	**Taxane**		5	(4.0)	1	(1.5)	4	(6.9)	
	**Anthracycline + Taxane**		116	(92.8)	64	(95.5)	52	(89.7)	0.232
**HER2 targeted**	**Trastuzumab**		28	(74.3)	6	(54.5)	20	(80.0)	0.116

**Table 3 cancers-17-02214-t003:** Mean and median mammographic percent density by response and survival outcomes.

		Frequency of Outcome				
		No		Yes		*p*
Outcome		*n*	%	*n*	%	*t*-Test
**cCR**		55	(50.9)	53	(49.1)	
Mean density		22.2		16.8		0.048
Median density		17.9		14.1		
IQR		5.6–37.4		3.7–24.2		
**pCR**		100	(78.9)	27	(21.1)	
Mean density		19.2		17.9		0.375
Median density		15.4		9.9		
IQR		5.5–29.6		3.4–31.4		
**Relapse**		84	(66.1)	43	(33.9)	
Mean density		17.1		22.4		0.041
Median density		13.5		22		
IQR		3.6–23.6		8.1–35.9		
**Distant relapse**		90	(70.9)	37	(29.1)	
Mean density		17.6		22.1		0.071
Median density		13.5		22.3		
IQR		4.4–24.3		6.9–35.7		
**Overall mortality**		79	(62.2)	48	(37.8)	
Mean density		18.4		19.8		0.324
Median density		14.5		18		
IQR		4.5–29.2		5.5–33.5		
**BrCa-specific mortality**		91	(71.7)	36	(28.3)	
Mean density		18		21.2		0.144
Median density		14.1		20.8		
IQR		4.2–29.2		6.6–33.5		

**Table 4 cancers-17-02214-t004:** Response and survival outcomes in high and low percent breast density patients.

	Density Category				
	Low Density		High Density		*p*
Outcome	*n*	%	*n*	%	Chi sq
**cCR**					
Yes	31	(58.5)	22	(40.0)	0.027
No	22	(41.5)	33	(60.0)	
**pCR**					
Yes	16	(25.0)	11	(17.5)	0.15
No	48	(75.0)	52	(82.5)	
**Any Relapse**					
Yes	19	(29.7)	24	(38.1)	0.158
No	45	(70.3)	39	(61.9)	
**Distant relapse**					
Yes	16	(25.0)	21	(33.3)	0.151
No	48	(75.0)	42	(66.7)	
**Overall mortality**					
Yes	25	(39.1)	23	(36.5)	0.383
No	39	(61.9)	40	(63.5)	
**BrCa specific mortality**					
Yes	16	(25.0)	20	(31.7)	0.2
No	48	(75.0)	43	(68.3)	

## Data Availability

The raw data supporting the conclusions of this article will be made available by the authors on request.

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
