# Peer review of "Correlations Between Mammographic Breast Density and Outcomes After Neoadjuvant Chemotherapy in Patients with Locally Advanced Breast Cancer"

_cancers, 2025, doi:10.3390/cancers17132214_

Round 1

Reviewer 1 Report

Comments and Suggestions for Authors

The paper explores the correlation between mammographic breast density (MBD) and both clinical and pathological complete response (c/p CR), as well as long-term outcomes following neoadjuvant chemotherapy (NAC), in a cohort of 127 selected patients treated at Royal Perth Hospital between January 2003 and December 2014. MBDs were assessed by Cumulus software for all mammograms of contralateral breasts in the cranio-caudal view, including both digitized film and digital formats.  

The authors found that by dividing the population at the median percentage dense area (PDA) into high and low MBD groups - cCR was significantly lower in the high MBD group, while pCR, relapse-free survival (RFS), and breast cancer-specific survival (BCSS) were all numerically worse but did not reach statistical significance. Higher mean percentage dense area was significantly associated with inferior cRS and relapse-free, but not with pCR nor BCSS.

The manuscript is well-structured and clearly articulated, with up-to-date and appropriate references.

The authors identified the limitations and proposed appropriate strategies to address them.

Author Response

Reviewer 1 - Open Review

We sincerely appreciate your thoughtful review of our manuscript and your recognition of its structure and clarity. 

We note there were no comments for consideration.

Reviewer 2 Report

Comments and Suggestions for Authors

The article by Agarwal et al reported the relationship between high mammographic breast density and high relapse in breast cancer patients for the first time. Overall it’s well written, but a bit more information is needed to support this finding.

Here are some minor suggestions for consideration:

  1. I saw the author added detailed information of chemotherapy and neoadjuvant treatment in Table 2. Still may be a good idea to include a basic background of these treatment & drugs used for breast cancer in the introduction section.
  2. The author introduced the concepts of pathological complete response (pCR) and clinical complete response (cCR). What’s the difference between them?
  3. In section 2.2 line 133-135 on page 4, is there a figure that could illustrate the threshold matching between film and digital mammography cohorts?
  4. In section 3.2.3 on page 8, the author found that obese patients with high MBD had worse outcomes than patients with lower BMI. When making such comparison, did the author rule out any potential bias between these two patient groups? Like patients might be at different tumor stages, higher stage could lead to worse prognosis by its nature regardless of the treatment. Patients might get different chemotherapy cycles and chemo drugs. Additionally, obese patient could have higher possibility of suffering from chronical diseases other than cancer itself, and that could bias the comparison. It might be useful to summarize all the clinical information, treatment and response parameters between obese and nonobese patients into tables.

Author Response

We thank the reviewer for their time, expertise and careful consideration of our manuscript. Please note we have highlighted changed text in the manuscript in red.

Comment 1: I saw the author added detailed information of chemotherapy and neoadjuvant treatment in Table 2. Still may be a good idea to include a basic background of these treatment & drugs used for breast cancer in the introduction section.

Response 1: Thank you for the suggestion – we have added a brief explanation of the core chemotherapy agents employed in neoadjuvant breast cancer chemotherapy in the introduction – line 61-63.

Comment 2: The author introduced the concepts of pathological complete response (pCR) and clinical complete response (cCR). What’s the difference between them?

Response 2: pCR is defined in the introduction – lines 66-68.  We have adjusted the description to make it clear it is the microscopic absence of invasive tumour in the surgical specimen.

cCR is defined in methods, lines 123-124. We have expanded the explanation of clinical tumour assessments and the clinical response outcomes clinical complete response (cCR) and clinical partial response (cPR).

Comment 3: In section 2.2 line 133-135 on page 4, is there a figure that could illustrate the threshold matching between film and digital mammography cohorts?

Response 3: We could not create a figure that we felt enhanced the explanation. We have therefore restructured the paragraph describing the method of creating high and low density cohorts for comparison to better convey the process, lines 143-151

Comment 4: In section 3.2.3 on page 8, the author found that obese patients with high MBD had worse outcomes than patients with lower BMI. When making such comparison, did the author rule out any potential bias between these two patient groups? Like patients might be at different tumor stages, higher stage could lead to worse prognosis by its nature regardless of the treatment. Patients might get different chemotherapy cycles and chemo drugs. Additionally, obese patient could have higher possibility of suffering from chronical diseases other than cancer itself, and that could bias the comparison. It might be useful to summarize all the clinical information, treatment and response parameters between obese and nonobese patients into tables.

Response 4: We now provide a supplementary table, supp table 1, at the end of the manuscript which details features of tumours by intersecting MBD and BMI categories. Comparing the patients with high MBD and high BMI to all others, notably the former have statistically more T3-4 tumours and numerically more involved nodes but numerically less high grade cancers. Patients with high MBD regardless of BMI are also younger, in keeping with the correlation of high MBD and younger age. These observations are also noted in the text which refers to the table; lines 278-283 in results and 375 to 378 in discussion.

Reviewer 3 Report

Comments and Suggestions for Authors

This study looks at how mammographic breast density (MBD) relates to outcomes after neoadjuvant chemotherapy in early breast cancer. It finds that higher MBD increases the risk of relapse but does not change the chances of achieving a pathologic complete response (pCR). In obese patients, high MBD is linked to lower pCR rates and may lead to higher relapse and breast cancer mortality, indicating a need for more research on how obesity and MBD interact with chemoresistance. The following are some major and minor suggestions to incorporate into the revised manuscript.

Minor Comments:

  • Tables 1-3 are very confusing. Restructure clearly.
  • Keep the same font-size for captions and axes in Figure-2.
  • Part-b of Figure 5 is blurry. Enhance its contrast.
  • Authors are suggested to insert a relevant image after the first paragraph of the Introduction section.
  • Pay attention to math-related items, such as 30 kg/m2. It is square of m.
  • Use clear symbol, e.g., >/< can be replaced with .

Major Comments and/or suggestion:

  • Elaborate conclusion section as it is relatively short and does not communicate the purpose of study clearly.
  • Elaborate section-3, it is not clearly written. Include more statistical charts/plots to enhance its readability.
  • Also, include some typical plots/images in Section-4.
  • The study reports that mean MBD is higher in relapsing patients but it did not vary by pCR or breast cancer deaths. How do the authors interpret these findings, and what implications do they have for clinical practice?
  • Why do the authors state that links between high MBD and longer-term outcomes remain uncertain? What factors could contribute to this uncertainty? Elaborate this fact in the discussion section more clearly.

Author Response

Many thanks for your expert consideration of our manuscript and helpful suggestion to improve it. Please note we have highlighted changed text in the manuscript in red.

Comment 1: Tables 1-3 are very confusing. Restructure clearly.

Response 1: Tables 1-3 have been recreated in excel and then imported to word to provide more clarity. Original excel tables will also be uploaded

Comment 2: Keep the same font-size for captions and axes in Figure-2.

Response 2: Figure has been adjusted to equalize the fonts.

Comment 3: Part-b of Figure 5 is blurry. Enhance its contrast.

Response 3: We have sharpened and increased the font size for figure 5a and 5b.

Comment 4: Authors are suggested to insert a relevant image after the first paragraph of the Introduction section.

Response 4: We are not entirely sure what type of figure the reviewer had in mind. As the first paragraph describes other roles of breast density we have created an outline of a new figure 1 summarizing how breast density may guide various aspects of breast cancer clinical care. However, this type of figure would be more common in review articles rather than original research based manuscripts.
If suitable we are happy to add detail and references and include. Note we have placed the figure currently at the end of the manuscript – lines 562-564

Comment 5: Pay attention to math-related items, such as 30 kg/m2. It is square of m.

Response 5: Thank you. The power has been superscripted to give the correct term - kg/m2. Table 1 also has this unit now included in the BMI data.

Comment 6: Use clear symbol, e.g., >/< can be replaced with .

Response 6: This symbol has been replaced by alternate text.

Comment 7: Elaborate conclusion section as it is relatively short and does not communicate the purpose of study clearly.

Response 7: The conclusion has now been expanded to include a restatement of purpose and how the results pertain to this. Lines 390-399

Comment 8: Elaborate section-3, it is not clearly written. Include more statistical charts/plots to enhance its readability.

Response 8: We are not sure as to the specific areas the reviewer has in mind in section 3 as this is the results chapter and so the core of the paper. We have added a supplementary table 1 to show the demographics of the 4 separate groups categorized by high/low BMI and high/low.

We were at our maximum allowed word count for the manuscript as a whole and so a general elaboration throughout the results section would be difficult but we can doubtless address specific areas needing attention and pare down words elsewhere. Could the reviewer specify please?

Comment 9: Also, include some typical plots/images in Section-4.

Response 9: As section 4 is the discussion section, plots and images would be unusual, usually being the province of the results section – again we would appreciate it if the reviewer could specify plots/images they would like us to add.

Comment 10: The study reports that mean MBD is higher in relapsing patients but it did not MBD vary by pCR or breast cancer deaths. How do the authors interpret these findings, and what implications do they have for clinical practice?

Response 10: Thank you, we agree more speculation of the underlying cause of our results is warranted. We have added a discussion paragraph on this early on in the discussion section. ‘The lack of statistical significance in some measures could attribute to insufficient events (cCR events and relapse events are higher than pCR and breast cancer death events). Alternatively, it could be that high MBD, and the underlying stroma this represents, reduce the ability of chemotherapy to kill bulk cell populations relative to low MBD lowering cCR but not to erradicate all cancer clones in responders leading to a similar pCR. Additionally, as cCR is a clinical measure, high MBD may reduce the clinicians ability to confirm disappearance of a tumour leading to a spuriously low cCR.’ Lines 304-310

We have also added a shorter discussion on the categorical results. Lines 313-316.

Comment 11: Why do the authors state that links between high MBD and longer-term outcomes remain uncertain? What factors could contribute to this uncertainty? Elaborate this fact in the discussion section more clearly.

Response 11: This statement has been removed and replaced by more detailed discussion.
The core reason we express uncertainty is that our cohort is relatively small and our longer term outcomes of borderline significance. We are however expanding our case numbers prospectively (as stated in the discussion – line 383) to address this and validate or refute our findings.

Reviewer 4 Report

Comments and Suggestions for Authors

The manuscript describes everything well, but needs some revision before acceptance. 

@ Definition of cPR deviates from standard criteria (e.g., RECIST), a clear justification or reference is required.

@ The exclusion criteria mention “eligible contralateral mammograms,” but it’s not clear what defines eligibility. Clarifying this earlier would improve transparency.

@ There is a large number of patients excluded due to unavailable or non-assessable mammograms. The potential impact of this missing data on study findings should be discussed.

@ Splitting breast density into high and low groups based on median values within each imaging cohort seems arbitrary. Considering breast density as a continuous variable or using clinically established cutoffs might provide more meaningful insights.

@ There is a notable difference in HER2-positive cases between the film and digital mammogram groups, which could confound the results. Adjusting for this imbalance in the analysis or performing stratified analyses would be advisable.

@ I recommend that the authors strengthen their manuscript by incorporating relevant recent literature that addresses related methodologies and findings in breast cancer diagnosis, treatment, and prognostic markers. Including up-to-date references will improve the scientific context and rigor of the study.  I strongly encourage the authors to conduct a thorough, up-to-date literature search using platforms such as Google Scholar to identify and include other pertinent studies. Upon search, I have got few relevant references which can be used if authors want. 

  • 10.1016/j.saa.2022.122000

  • 10.3788/COL202018.051701

  • 10.1002/pon.70078

  • 10.2147/IJN.S466042

  • 10.1245/s10434-024-16454-8

  • 10.1016/j.jep.2024.119126

  • 10.25259/Cytojournal_127_2024

  • 10.25259/Cytojournal_37_2023

  • 10.1007/s40005-025-00731-z

Author Response

Comment 1:  Definition of cPR deviates from standard criteria (e.g., RECIST), a clear justification or reference is required.

Response 1: We have emphasized more clearly that this is an examination assessment and expanded the description of the assessment. We have also referenced the assessment which is based on the older WHO criteria. We did not use RECIST as around 95% of cases showed a RECIST response compared to 78% with WHO based analysis. Lines 123-128

Comment 2: The exclusion criteria mention “eligible contralateral mammograms,” but it’s not clear what defines eligibility. Clarifying this earlier would improve transparency.

Response 2: Reference lines 100 to 109 give details of patient eligibility in terms of tumour stage and treatment. This paragraph also references the mammogram eligibility requirements that come subsequently. We have added detail in the mammography section of methods, namely ‘Patients were considered ineligible if CC view mammograms were missing, post-surgical, of the ipsilateral breast, or of insufficient image quality for reliable density assessment.’ Lines 137-139

Comment 3: There is a large number of patients excluded due to unavailable or non-assessable mammograms. The potential impact of this missing data on study findings should be discussed.

Response 3: A good point. We have added comment on this point (lines 157-161) regarding patients with and without available mammograms, ‘For the 111 ineligible patients, more patients were from the pre-2008 film mammography group (n=63) than the 2008-onward digital mammogram group (n=48) due to older and physically archived images being more likely to be untraceable than digital images. However, no clinically or statistically significant differences in personal and tumour demographics or treatment were observed for these excluded patients.

Comment 4: Splitting breast density into high and low groups based on median values within each imaging cohort seems arbitrary. Considering breast density as a continuous variable or using clinically established cutoffs might provide more meaningful insights.

Response 4: We did analyze breast density as a continuous variable, see line 218 onwards and table 3, ‘Considering the associations of MBD as a continuous variable and BC outcomes ……’.

Considering  assessing the impact of breast density as a categorical variable, this is required in order to apply findings to patients. To use data clinically, a patient needs to be assigned to a category with known clinical associations that then assist in management. The most common comparison of MBD by outcome is to use the BIRADs scheme which broadly aims to divide patients into quartiles A-D with high density often characterized by Birads C-D, comparable to dividing at the median. We also used the median in particular as we assessed this as the best way to combine film and digital cohorts where the BD varied between the two.

We have added a more detailed explanation of the rationale for using the median ‘The median was chosen as a cut-point, as this reflects current clinical assessments whereby the BI-RADs MBD scoring system system designates the upper two categories, heterogeneously dense and extremely dense breasts as high breast density which corresponds to approximately 50% of women in our region.’ Lines 148-151

Comment 5: There is a notable difference in HER2-positive cases between the film and digital mammogram groups, which could confound the results. Adjusting for this imbalance in the analysis or performing stratified analyses would be advisable.

Response 5: We do already acknowledge this difference in the text (line 165) but have added an explanatory note and agree that cross-checking this as a confounder is warranted. We have added ‘line 166 – 169) This likely reflects the shift in protocols and practice from selecting patients for NAC based on primary tumour size in the earlier cohort to selection based on tumour biology in the later cohort. In more recent years NAC has been routinely incorporated into HER2-positive early breast cancer  for all but small tumours.’

Although numbers do not permit multi-variate analysis we have also cross-checked for influence of tumour biology. We have added at line 241-244 , ‘Considering potential confounders, there was no difference between tumour demographics or biology between the high and low mammographic density cohorts. In particular, HER2 status, the biggest differential between film and digital mammography groups did not vary with 30.6 and 29.2% of patients in low and high breast density breast groups being HER2 positive.’

Finally on this point we have added in the ‘study weaknesses’ section, line 374 381 ‘Weaknesses include the relatively low case number which precluded analysis of the predictive import of MBD in breast cancers of differing biological sub-type. Additionally, the low rate of use of HER2 targeted agents in the HER2 positive cohort makes the relevance of these results to contemporary HER2 positive patients where such targeting is routine uncertain.

Comment 6: I recommend that the authors strengthen their manuscript by incorporating relevant recent literature that addresses related methodologies and findings in breast cancer diagnosis, treatment, and prognostic markers. Including up-to-date references will improve the scientific context and rigor of the study.  I strongly encourage the authors to conduct a thorough, up-to-date literature search using platforms such as Google Scholar to identify and include other pertinent studies. Upon search, I have got few relevant references which can be used if authors want. 

Response 6: these articles appear not to relate to predictive biomarkers of chemotherapy or intersect with mammographic density. We do not believe (with our limited word count) that extensively summarizing basic aspects of breast cancer care not specifically related to predictive biomarkers for chemotherapy or aspects of mammographic density will enhance the manuscript.

Specifically looking at the articles mentioned:

https://doi.10.1016/j.saa.2022.122000 - I am unclear as to the relevance of this article to the subject material of our article. It describes an experimental model of breast cancer sub-type diagnosis which is by no means standard of care anywhere. Our manuscript in no way relates to diagnosis but to prediction and prognosis.

https://doi.10.3788/COL202018.05170110.1002/pon.70078 - again this article relates entirely to diagnostic technique and so does not intersect with the subject of our manuscript in any way beyond it relates to breast cancer.

https://doi.10.2147/IJN.S466042 - this article describes development of a nano-biomimetic delivery system and focuses on cervical cancer with the model being HeLa cells (a cervical cancer line). The only connection I can see is in that this system is purported to address treatment in hypoxic tissues. It is possible that high density breast tissue may be more hypoxic. However the first and senior author have already published a review of the role and management of hypoxia in cancer but we consider it too far removed to cite in this paper - Hypoxia as a signal for prison breakout in cancer. Andrew Redfern, Veenoo Agarwal, Erik W Thompson. DOI: 10.1097/MCO.0000000000000577

https://doi.10.1245/s10434-024-16454-8 - we are familiar with the systemic immune-inflammation index (SII) described in this paper which is one of many hundreds of papers looking at a variety of immune based and other prognostic signatures. Work by our collaborators on the underlying biology of MBD does demonstrate altered immune infiltrates between high and low density tissue. We would not however, consider this to be of close enough relevance to cite.

https://doi.10.1016/j.jep.2024.119126 - this study looks at a herbal remedy being used to treat cervical cancer. I can see no link to our work.

https://doi.10.25259/Cytojournal_127_2024 - this article is slightly closer, being an analysis of on-line datasets, looking for prognostic transcription factors in TNBC. However, the work does not access treatment data so does not identify predictive factors which is the area of our work in chemoresistance.

https://doi.10.25259/Cytojournal_37_2023 - this article is about the intra-operative use of imprint cytology to predict eventual LN status. We do not believe it is at all relevant to our work.

https://doi.10.1007/s40005-025-00731-z - this article is slightly closer, relating to novel chemotherapy formulations but the work does not relate to breast density.

Round 2

Reviewer 4 Report

Comments and Suggestions for Authors

Thanks for addressing the concerns!